# Wellbeing, Sense of Belonging, Resilience, and Academic Buoyancy Impacts of Education Outside the Classroom: An Australian Case Study

**DOI:** 10.3390/bs15081010

**Published:** 2025-07-25

**Authors:** Helen Cooper, Tonia Gray, Jacqueline Ullman, Christina Curry

**Affiliations:** 1School of Education and the Centre of Educational Research, Western Sydney University, Penrith 2751, Australia; j.ullman@westernsydney.edu.au (J.U.); c.curry@westernsydney.edu.au (C.C.); 2World Leisure Centre for Excellence (WLCE), Western Sydney University, Penrith 2751, Australia

**Keywords:** student health and wellbeing, outdoor education, outdoor learning, school belonging, secondary school

## Abstract

This paper examines the importance of ‘education outside the classroom’ (EOtC) in an Australian secondary school. The primary aim was to develop a sense of belonging, build resilience, and enhance wellbeing in female students. This study investigated two cohorts of Year 9 students (aged 14–15 yrs) who participated in a four-week residential EOtC pilot program. The first cohort (Wave 1; *N* = 58) undertook the program alongside (*N* = 39) boys. The second cohort was single-sex girls (Wave 2; *N* = 28). A mixed-methods research design was implemented to inform experiences of students, parents, and staff and to triangulate inferences drawn from the data. Quantitative data was gained from pre- and post-program surveys with students and parents, whilst qualitative data was gathered from student focus groups, staff, and parents through semi-structured interviews to assess more nuanced impacts. School belonging was measured using the PISA six-item scale. Academic buoyancy was quantified using the four-item Academic Buoyancy Scale. Self-efficacy, peer relations, and resilience were evaluated by employing the 34-item Adolescent Girls’ Resilience Scale. The findings revealed significant improvements in students’ sense of belonging, including higher levels of school belonging than reported Australia-wide averages for 15-year-olds. Despite students’ mean academic buoyancy scores being more than a point lower than reported baseline scores for Australian high school students, it was promising to see a modest increase following the EOtC program. In conclusion, EOtC is a potent vehicle for developing a sense of belonging, enhancing resilience, and equipping students to deal with academic challenges.

## 1. Introduction

The Australian education system is not thriving, as indicated by teacher workforce shortages ([5]), stressed school leaders ([67]) and teacher burnout ([36]). Globally, the system is at breaking point ([66]). Increasing levels of anxiety and depression reveal that Australian students are not thriving ([2]), as evidenced by declining academic standards ([3]) and increases in problem behaviours in classrooms ([56]). The disciplinary climate in Australia’s classrooms has deteriorated to such an extent that a Senate inquiry into the disruption of student learning has been announced ([7]). The latest Programme for International Student Achievement (PISA) results rank Australia 10th out of 81 countries for the most frequently bullied students ([57]), with 18% of students reporting loneliness and 21% feeling left out at school ([58]). For this reason, educational reform is seeking alternatives to traditional classroom learning. Against this backdrop, there has been momentum to rethink the curriculum to embrace wellbeing and a sense of belonging as key indicators of school success ([11]; [66]).

### 1.1. Nature and the Outdoors as Teacher

Research is amassing to indicate that outdoor learning environments provide unique opportunities for personal growth and development ([44]; [61]). The outdoors is a unique learning environment that cannot be replicated in an indoor classroom setting ([60]). Education outside the classroom (EOtC) provides young people with an opportunity to be physically active while learning and developing a relationship with the natural world ([30]; [31]; [49]). Teaching in partnership with the natural world provides a compelling platform for transformational learning ([28]; [32]; [54]), particularly for adolescent girls ([6]; [27]; [51]; [76]). Additionally, the outdoor learning environment has grown in popularity and has repeatedly been shown in schools to be a potent vehicle to amplify 21st-century competencies ([8]; [23]; [43]; [44]; [46]; [45]).

Given this educational milieu, there is a growing body of evidence that nature can have a positive impact on mood, coping skills, and personal development ([39]). Nature-based experiences restore psychological deficits and promote cognitive development ([42]; [63]; [71]). Globally, EOtC activities are being implemented in schools to promote wellbeing ([12]; [24]; [25]; [65]) and expand 21st-century skills which are linked to better life outcomes, including academic success, greater life satisfaction, healthier behaviours, less test and class anxiety, and more ambitious career plans ([59]). [39] ([39]) present evidence that lessons in nature provide greater student engagement in subsequent lessons, in addition to the benefits directly attributable to their EOtC experience.

EOtC is grounded in constructivist theories of education; that is, the learner is an active participant in the acquisition of knowledge. [20] ([20]) mentions in his foundational text Education and Democracy, “as formal teaching and training grow in extent, there is the danger of creating an undesirable split between the experience gained in more direct associations and what is acquired in school” (p. 7). Dewey’s prescience is becoming more evident as technology advances, and students seem to have little or no direct contact with the subject matter they are endeavouring to learn.

### 1.2. EOtC Impacts and the Residential Model

For this research project, [35] ([35]) define resilience as “the ability to react to adversity and challenge in an adaptive and productive way” (p. 40). Resilience skills are a component of healthy development for youth ([24]), and developing protective coping skills allows young people to deal with future adversity ([35]). Protective factors are the specific psycho-social strengths that people possess that allow them to combat adversity ([68]). Protective factors associated with resilience include an extensive array of individual attributes and external supports. Individual assets examined in this study are (1) sense of belonging, (2) academic buoyancy, and (3) resilience. Although many factors are external to the individual and harder to develop, substantial support has been offered that youth EOtC programs can foster the development of resilience through effective programming ([24]; [35]).

Additionally, [15] ([15]) stated that residential EOtC experiences provide opportunities and benefits that cannot be achieved in any other educational context or setting. Having an EOtC program take place during Year 9 is also significant. Notably, this is a time in adolescent lives where issues such as increasing use of technology, initiation of drug and alcohol use, sexual abuse, and family breakdown are faced, contributing to a decline in mental health and wellbeing ([55]). Furthermore, [34] ([34]) espouses the great rewiring of childhood and increased digital usage, which has led to an epidemic of mental illness or “Generation Overwhelmed”.

Australian research ([1]) consistently demonstrates an upward trend in levels of psychological distress among young females. Mental health is a key contributor to general health and wellbeing ([4]; [78]), with 14% of children aged 4–17 in Australia estimated to have experienced a mental illness in the past 12 months ([1]). Poor mental health can impact the potential of young people to live fulfilling and productive lives ([4]). Increases in psychological distress were seen particularly for 18–24-year-olds in Australia, with the proportion of people experiencing elevated levels of psychological distress increasing from 14% in 2017 to 22% in April 2020 ([4]).

### 1.3. Girls’ Participation in EOtC in Australia

Although depression and anxiety affect people of all ages, the risk of developing mental disorders is more likely in youth, which can have a persistent impact throughout the lifespan ([48]). Girls are particularly sensitive to external influences such as peer interactions ([53]). [69] ([69]), the author of several books on girl culture, writes, “We live in the age of the fiercely successful ‘amazing girl’” (p. 2) with girls outshining their male counterparts in academics and leadership roles. However, these advancements are at a psychological cost that [69] ([69]) labels ‘The Curse of the Good Girl’. This phenomenon has been shown to diminish girls’ resilience or ability to cope with stress ([69]). Seeking social support and accepting social resources assists girls in coping with daily stressors ([17]). 

[29] ([29]) refers to the traditional approach to outdoor education in Australia as a “boys club, built on a masculinised Outward Bound Model” (p. 41), where gender inequality abounds. Historically, this approach has prioritised activities such as hiking, rock climbing, and high-ropes courses, which have been perceived as masculine and ‘hard-skills’ based ([10]). As a result, there has been an underrepresentation of women and other marginalised groups in EOtC and a lack of diversity in the types of activities offered. Inadequate representation has also occurred in staffing, particularly in outdoor education leadership positions, which have been dominated by older men ([29]).

In recent years, there has been a push towards a more inclusive approach to outdoor education, with a focus on promoting diversity and representing women in the types of program offerings and leadership positions ([41]; [52]). Girls who participated in EOtC programs improved resilience in the form of increased sense of mastery, increased sense of relatedness and less emotional reactivity ([77]). A recent study by [50] ([50]) discusses young women who shared how their identities and wellbeing were profoundly influenced by their experiences in adolescent outdoor learning, which continue to resonate in their lives. Four key narratives emerged from their accounts: (1) the nourishing effects of nature; (2) the cumulative benefits of critical environmental approaches and simplicity; (3) the importance of structured support alongside learning independence; and (4) the development of a strong female identity, highlighting the lasting impact of immersive outdoor experiences on their wellbeing ([50]).

There are several long-established outdoor EOtC programs in Australia, such as Geelong Grammar School’s ‘Timbertop’ ([26]) and The Scots College ‘Glengarry’ Program ([45]). Similarly, these programs operate from a regional campus outside the main school, with traditional teaching integrated with EOtC. The study school has developed an innovative EOtC program for girls that breaks some of the stereotypes and expands the offerings to best suit the population of girls. This research represents one of the first of its kind in the Australian school context for an all-girl setting. It provides meaningful, robust results that add to the existing knowledge in the field of outdoor education. The researchers systematically examined whether girls’ levels of resilience, sense of belonging, and academic buoyancy increased post-participation compared to pre-participation.

### 1.4. The Context: EOtC at the Study School

The Year 9 program is the cornerstone of the study school EOtC continuum that is embedded throughout the school years from Kindergarten (5–6 years old) to Year 12 (17–18 years old). Female students in Year 9 at the study school are typically 14–15 years old. The Year 9 four-week residential EOtC program was designed with a mix of expeditions, academics and skill development. The EOtC campus and expedition locations are situated in spectacular, dynamic landscapes where students are immersed in the natural environment. The program is not a ‘survival’ type program, but one that facilitates connections with the natural world safely, through staging and sequencing outdoor experiences and expeditions with the comforts of ‘home’. The site was purchased by the school following significant evidence of poor student wellbeing that had impacted the school community. Following this, the school redefined their approach by offering a radical alternative educational setting to support student wellbeing and build resilience. 

The EOtC campus is a more informal setting where no specific uniform requirements are implemented. This is in direct contrast to their regular school, where a strict uniform code is applied. The outdoor campus offers space for personal as well as academic pursuits, so the boundaries between personal space and the learning environment become blurred. Student accommodation was in a shared room in a relatively luxuriously appointed lodge with a private ensuite for each room. Breakfast was eaten in the large common area in the lodge, lunch was packed at breakfast and eaten outdoors, and dinner was cooked by professionals and served in a communal dining hall. Students were expected to assist with some cleaning and serving duties, as well as do their own clothes washing. 

The two four-week pilot programs were structured as follows:2 × 3-day hiking expedition;Several academic learning blocks, including one major project/presentation;Abseiling, high ropes course and other outdoor activities on-site at the EOtC campus; Several experiential learning activities;Service-learning day at neighbouring property;Five-day expedition trekking back to the main school campus.
On-site outdoor education facilities include bushwalking, canoeing and raft building, rock climbing and abseiling, high and low ropes, mountain biking, team challenges and zip-lining. 

Two four-week pilot programs were conducted at the study school in 2022; one was co-educational, and one was girls-only. Extant research discusses co-educational and single-sex outdoor programs, particularly around the benefits for girls ([10]; [74]; [75]). Various literature reports that girls would opt out rather than show up as more capable than boys across a whole raft of subjects ([13]; [72]). The decision to include boys in one of the programs in this study was brought about by many competing elements, such as exploring the differential wellbeing and resilience outcomes of co-educational models. 

Both programs were designed to ensure students did not have access to their phones or social media for the duration, as it has been widely reported that digital devices have deleterious impacts on student wellbeing; see, for example ([40]; [70]). Ownership and usage by teenagers has escalated with social media uptake, with apps such as Instagram, TikTok and Snapchat being used widely by students in 2022. [70] ([70]) concluded that 23.3% of children and young people reported use at problematic levels, which is associated with increased risk of depression, increased anxiety, higher perceived stress, and poorer sleep quality.

### 1.5. Research Objectives

This paper centres on outcomes for female students attending an EOtC program as part of a Year 9 experience at an elite all-girls school whose main campus is located in a metropolitan region of Sydney, Australia. The school is a high socio-economic, largely Anglo, academically competitive private school in an affluent suburb of Sydney. A pilot program was undertaken in 2022 to establish baseline data and inform program design. The program was based at the EOtC campus, with three journeys taking place off-site and culminating with the arrival on foot back to the main school campus.

The research objectives and associated parameters were determined in consultation with the school. The articulated objectives were to

Assess the impact of the Year 9 residential pilot program, through the perspectives of students, parents and staff;

Inform future programming needs for students;

Create a framework for further program evaluation; 

Determine the scope for potential further research.

In 2022, two student cohorts participated in the four-week pilot program. The first cohort consisted of 58 girls who participated in the program from August to September 2022 alongside 39 boys from a private boys’ school of similar socio-economic status. The second cohort comprised 28 girls who participated in a single-sex cohort from October to November 2022. These two cohorts constituted the pilot program, as the findings from this study were presented to the school following data analysis to refine the program for ongoing cohorts that have continued to participate in the proceeding years.

## 2. Methodology

The research project set out to identify if students’ sense of belonging, academic buoyancy and resilience were enhanced due to engagement in the study school’s residential EOtC program. These factors were determined to be integral, especially in terms of wellbeing indicators associated with the school’s desire to develop courage and resilience and to decrease or mitigate stress in their students. A mixed-methods research (MMR) design was adopted by the research team, utilising both quantitative and qualitative data sets to inform the experiences of students, parents and key stakeholders to triangulate the inferences drawn from either data set, thus strengthening the research ([9]; [18]). Survey methodology and instrumentation were determined by a panel of experts from Western Sydney University and the outdoor education field, in collaboration with the study school. This study was granted human ethics approval H14564.

A series of surveys, interviews and focus groups were conducted in 2022 with school executives and staff, students and parents. The data was analysed to determine the impacts and experiences of young people undertaking a four-week outdoor education program. The research questions that guided the research included the following:Do students report a greater sense of belonging after their engagement in their residential EOtC program?Do students report feeling more equipped to deal with academic setbacks and challenges after their engagement in their residential EOtC program?Do students report greater capacity for resilience after their engagement in their residential EOtC program?

Table 1 outlines the data collection methods employed in the study. All Wave 1 and Wave 2 students were invited to participate based on their enrolment at the study school. Not all data collected is displayed in this paper due to the need to retain anonymity and confidentiality of the study school.

### 2.1. Quantitative (Survey) Methodology

Two online surveys were completed by students and parents in the two programs (referred to throughout as “Wave 1” and “Wave 2”). Students and parents completed one survey before departure and another upon return.

#### 2.1.1. Student Survey Instrumentation

The student survey collected a small set of demographic items. The core constructs under investigation were measured using previously validated measures aligned with the central outcomes of the program. These included a sense of belonging, academic buoyancy, and resilience.

##### Sense of Belonging

A student’s sense of school belonging and isolation was measured using the Programme for International Student Assessment (PISA) 6-item scale ([19]). Sense of belonging relates to ‘feelings of being accepted and valued by their peers, and by others at their school’ ([79]). Students were asked to rate their agreement on a four-point Likert scale from ‘strongly agree’ to strongly disagree. Higher scores indicate a greater sense of belonging. Results of this can be compared to international (PISA) scores.

##### Academic Buoyancy

A student’s academic buoyancy was measured by the 4-item, psychometrically validated Academic Buoyancy Scale ([47]). Academic buoyancy is defined as ‘a student’s ability to successfully deal with academic setbacks and challenges that are typical of the ordinary course of school life’ ([47]). This scale specifically addresses students’ problem-focused coping in response to everyday hassles, stressors and strains ([47]). Students rate themselves on a 1 to 7 Likert scale, from ‘strongly disagree’ to ‘agree strongly’ to the four items.

##### Resilience

A female student’s capacity for relationship building, self-confidence, self-efficacy beliefs, and their approaches to challenges were measured using the 34-item Adolescent Girls’ Resilience Scale ([75]). This scale assesses the ‘degree to which girls’ beliefs about how they are able to solve problems, cope with stress, be brave and courageous, and persist when things get hard’ ([75]). Responses are given on a 1 to 5 Likert scale, from ‘strongly disagree’ to ‘strongly agree’.

#### 2.1.2. Parent Survey Instrumentation

Parents were surveyed at two time points: before their child departed for the program and shortly after their return. Parents were asked to rate a series of 26 possible anticipated outcomes of their child’s participation in the program against four possible outcomes:Not at all—I am not concerned about this area;Slightly—My child would benefit from minor changes in this area;Noticeable—I am hoping for a sustained change in this area for my child;Very much—I am hoping for a clear and definite change in this area for my child.

No personally identifying data was collected from parents, which would have allowed for survey matching across both periods. Accordingly, reported differences across the two surveys for parents reflect general trends rather than individual change(s). Together, these items were used to provide an overall sense of parents’ hopes and aspirations as related to the potential impact of the EOtC experience.

### 2.2. Qualitative Methodology

The quantitative surveys provided significant data, which was complemented by qualitative data to ensure authentic and candid responses from students, parents and staff. These included semi-structured interviews and focus groups performed during the program at several points in time (Table 1) that were captured to understand the nuances of experience.

The data gained was from interviews and focus groups undertaken either (1) on-site at the EOtC campus; (2) at various expedition sites; (3) back at the main school; and (4) via ZOOM (video conferencing). All interviews were audio recorded and transcribed. Audio transcriptions were categorised into key emerging themes by researchers into an Excel spreadsheet. Data was interpreted through the lens of constructivist grounded theory, which focuses on how participants co-construct the experience with the interviewer and allows for an iterative process to take place as successive data is captured ([14]).

#### 2.2.1. Focus Group Methodology [Students]

Focus groups were facilitated to spark conversation between students regarding their wellbeing, sense of belonging, and academic concerns during the program. One technique that was consistently used throughout focus groups was three words. This approach was selected to crystallise the essence of the student’s experience at that point in time. Focus groups were performed on an ad hoc basis around school programming and educational commitments.

#### 2.2.2. Interview Methodology [Students; Parents; Staff]

Semi-structured interviews were performed with students, parents and staff regarding wellbeing, sense of belonging, and academic concerns during the program. A small set of open-ended items included in interviews allowed respondents to share any questions, reservations, or other observations with researchers not included in interview questions.

## 3. Results and Discussion

### 3.1. Comparing Apples and Oranges

For a variety of reasons, for the inaugural residential experience at the EOtC campus, the study school decided to create a co-educational four-week Year 9 cohort. This was to trial the experience of girls mixing with boys of the same age and year level from a boys’ school of similar socio-economic background. The second four-week cohort comprised only girls. The 2022 programs were both four weeks in length and optional.

### 3.2. Wave 1: The Program—Co-Educational

In the first wave of students, fifty-eight girls (*N* = 58) had self-selected to be part of the first cohort, with the knowledge that this program was to be co-educational. Thirty-nine (*N =* 39) Year 9 boys arrived one week into the girls’ program.

#### 3.2.1. Student Survey Data

In Wave 1 of data collection, pre- and post-survey data were collected from *N* = 44 participating Year 9 female students. A completion rate of 79% for the two surveys and all student data reported from this wave is from this matched data set. Of this cohort, 11.4% (*n* = 5) indicated that they boarded at school, with their family homes located either in a rural (country/bush) location (*n* = 4) or in a regional (small town) location (*n* = 1). One participant (2.3% of the cohort) identified as a First Nations community member. While most participants reported only speaking English at home, 29.5% of the participating cohort (*n* = 13) spoke an additional language at home. Languages included Mandarin (11.4%), Arabic (2.3%), Japanese (4.5%), Cantonese (2.3%), Shanghainese (2.3%), German (2.3%) and Spanish (2.3%).

##### Adolescent Girls’ Resilience Scale (AGRS)

In Wave 1, students’ scores on all three subscales of the AGRS increased between pre- and post-survey deployment. Table 2 reveals results from a statistical comparison of mean scores at pre-/post-survey deployment. As evidenced by the post-experience increases in students’ “Approach to Challenges”, the subscale was large enough to be statistically significant [*t* (43) = −2.60, *p* = 0.006], while students’ mean score increase for the “Self-Efficacy” subscale was approaching significance [*t* (43) = −1.56, *p* = 0.063]. Cohen’s *d* effect sizes for both were in the ‘small’ range ([16]). All reported scores hovered between “neither agree nor disagree” (3) and “agree” (4), approaching the latter at the post-survey time period.

##### Sense of Belonging

As exhibited in Figure 1, students participating in Wave 1 demonstrated notably higher levels of school belonging than reported Australia-wide averages for 15-year-olds ([19]) across each of the six included items of the PISA Sense of Belonging measure. These are displayed by the overall percentage of dis/agreement (depending on the directionality of the item), with higher percentages indicating a greater sense of belonging at school. While differences between the pre-/post-survey iterations were not large enough to be statistically significant, it is notable that across all three items indicating experiences of isolation at school [e.g., “I feel like an outsider at school”; “I feel awkward and out of place in my school”; “I feel lonely at school”], students reported greater percentages of disagreement at the time of the post-survey.

##### Academic Buoyancy Scale

Finally, Figure 2 shows the differences in students’ academic buoyancy, or their capacity to bounce back from academically stressful setbacks at school across both pre- and post-survey iterations. While students’ mean academic buoyancy scores were more than a point lower than reported baseline scores for Australian high school students (*M* = 4.69, as reported in [47] ([47])), it is promising to see a minor rise in the scale mean across the two conditions. At both time points, students’ average scores hovered between “disagree somewhat” (3) and “neither agree nor disagree” (4).

#### 3.2.2. Parent Survey Data

In the pre-program survey, fourty-four (*N* = 44) parents provided useable data, including *n* = 36 mothers, *n* = 7 fathers, and one individual who did not provide this data. The post-program evaluation provided twenty-three (*N* = 23) responses, with *n =* 20 of these indicating that they had also participated in the pre-survey.

As depicted in Figure 3, parents were most hopeful about the impact participating in the program might have on their child’s use of technology, with nearly two-thirds of a point difference between this item and the second highest ranked item (‘expand[ing] their knowledge of the world’).

#### 3.2.3. Student Interview and Focus Group Data—Wave 1

Interview and focus group data were coded based on key themes that arose during the process of data collection, according to [14] ([14]) constructivist grounded theory principles. Key themes that emerged during data analysis of student interviews and focus group data include the following attributes and qualities: Resilience; Bravery; Absence of technology; Academic concerns; Wellbeing; Co-educational program dynamics. 

Parent and academic staff interview data have not been included as part of this paper. However, field staff interview data is presented alongside student data to support or refute claims in a later section of this paper. 

##### Three Words

At various times throughout the program, students were invited to provide three words that summed up their experience up to that moment. There were twenty-five (*n =* 25) responses from students in Wave 1 that have been used to create a word cloud as illustrated in Figure 4.

### 3.3. Wave 2: The Program—All Girls

The second group of twenty-eight (*N* = 28) students began their program following Wave 1. This cohort (and/or their parents) selected participation in Wave 2 solely because the program was all girls rather than its previous co-educational counterpart. The program elements were the same as Wave 1, with an equal number and duration of expeditions, academic time and skills development.

#### 3.3.1. Student Survey Data

In the second wave of survey deployment, *N* = 19 students who attended the program completed both the pre- and post-surveys, deployed before and at the end of their experience. This represents a completion rate of 68% for the two surveys. All student data reported from this wave is from this matched data set. Differences between Wave 1 and 2 are that the latter had no Boarders or First Nations participants. While most reported that they only spoke English at home (*n* = 12), 36.8% of Wave 2 (*n* = 7) spoke an additional language at home, including Mandarin (10.5%), Korean (10.5%), Hindi (5.3%), Japanese (5.3%), Romanian (5.3%) and Tamil (5.3%).

##### Adolescent Girls’ Resilience Scale (AGRS)

In Wave 2, students’ scores on all three subscales of the AGRS increased between pre- and post-survey deployment. Table 3 displays the results from a statistical comparison of mean scores at pre-/post-survey deployment. As can be seen, the post-experience increases in students’ “Approach to Challenges” and “Relationship Building” subscales were large enough to be approaching statistical significance [t(18) = −1.70, *p* = 0.053 and t(18) = −1.66, *p* = 0.057, respectively]. Cohen’s d effect sizes for both were in the ‘small’ range ([16]). For Wave 2, students’ mean scores on the “Self-Efficacy” subscale indicated a very minor decrease at post-survey deployment.

##### Sense of Belonging

As shown in Figure 5, students participating in Wave 2 exhibited higher levels of self-reported school belonging than reported Australia-wide averages for 15-year-olds ([19]) across four of the six included items of the PISA Sense of Belonging measure. Comparing the pre-survey and post-survey iterations, no differences were apparent for five of the six items in the measure. For the item “Other students seem to like me,” a six-percentage-point increase across the two time points was apparent for this cohort, rising from 89% at pre-survey to 95% at post-survey.

##### Academic Buoyancy Scale

Finally, Figure 6 uncovers the differences in students’ self-reported academic buoyancy. While students’ mean academic buoyancy scores at the time of the pre-survey were almost a point lower than reported baseline scores for Australian high school students (M = 4.69, as reported in ([47]), it is promising to see a slight rise in the scale mean across the two conditions. At the post-survey time point, participating students’ average scores increased to just above the neutral response option (“neither agree nor disagree” (4)).

#### 3.3.2. Parent Survey Data

During Wave 2 of survey data collection, parents were surveyed at a single time point prior to their child departing for the EOtC campus. In the pre-program interview, *N* = 19 parents provided useable survey data, including *n* = 16 mothers and *n* = 3 fathers. The majority of this cohort reported that they only spoke English at home (*n* = 12; 63.2%), with the largest subcohort of multi-/bi-lingual households reporting speaking Asian languages, including Mandarin, Cantonese and Korean.

As indicated in Figure 7, parents of Wave 2 were most hopeful about the impact participating in the program might have on their child’s development of leadership skills and their time away from technology.

#### 3.3.3. Student Interview and Focus Group Data—Wave 2

Following categorisation of interview and focus group data from Wave 1, additional student data from Wave 2 was categorised, supporting the themes developed in Wave 1. As this was an all-girls group, co-educational group dynamics did not emerge as a theme. However, the comparison of experience between Wave 1 and 2 did emerge due to the differences in overall group size and composition.

##### Three Words

Similar to Wave 1, at various times throughout the experience, students were asked to provide three words to describe how they were feeling about the program at that moment in time. There were thirty-three (*n* = 33) student responses from Wave 2 over this period that have been used to create a word cloud, as outlined in Figure 8.

#### 3.3.4. Insights from Qualitative Data

Through analysing and classifying qualitative interview and focus group data according to [14] ([14]) constructivist grounded theory, the key themes from participating students and field staff are summarised below: No technology: The absence of phones during the program was seen as beneficial by students and staff. Students acknowledged the positive impact on time management and expressed a willingness to improve their habits in the future.Academic concerns: Students perceived the academic aspect of the program as less effective compared to learning in a traditional school setting. They expressed concerns about missing vital content and highlighted the importance of having appropriate teachers for each subject.Co-educational program dynamics: While students reported positive experiences during the co-ed program, some staff members held different views. Staff noted that students in the co-ed cohort were often preoccupied with social dynamics, resulting in less focus on learning key skills.Downtime and program duration: Students from both waves expressed a need for more downtime, including one morning each week for sleeping in. Staff showed interest in extending the program’s duration.Gratitude and appreciation of privileges: Students from both waves expressed a new-found understanding of their privilege, and a sense of gratitude for their school and family.Impact on wellbeing attributes: Students’ sense of belonging showed the most significant improvement across both qualitative and quantitative data sets. Changes in bravery and mastery attributes were observed throughout the program.Resilience and interplay of wellbeing attributes: Participation in the EOtC program positively influenced students’ resilience. The interplay of belonging, bravery, and mastery had a profound impact on overall wellbeing.Cohort differences: It was noted that comparing students across Waves 1 and 2 was challenging due to differences, including co-educational and all-girls groups, varying numbers of students and groups, and different staffing arrangements. The two cohorts included in this study were fundamentally different, as outlined in Table 4.

### 3.4. Limitations of the Study

The researchers acknowledge there are limitations associated with this pilot study. Most notably, it did not include an indoor classroom control group. Controlling for teacher and teacher/context effects would also be useful in future research. In defence of this shortcoming, readers need to be aware that this was a small-scale pilot study with limited research funding attached. To redress this limitation, future studies should investigate this addition in greater depth.

Another limitation of this study was the fact that it was undertaken at an elite, single-sex private school. For this reason, the findings cannot be generalised to other populations that lack the financial resources to participate in extended EOtC programs. Finally, the Adolescent Girls Resilience Scale is only applicable to girls and cannot be applied to non-female students.

### 3.5. Summative Comments

The evaluation of the EOtC long-stay program revealed that students’ participation in the program facilitated further development of a strong sense of belonging, as indicated in quantitative findings. Students began with higher than the Australian average school belonging, which was further developed during the program. This is not surprising considering the high socio-economic status of participating students. Students consistently mentioned that they appreciated the opportunity to make friends from different friendship groups. This, in turn, fostered social capacity building and enhanced interpersonal relationships. Moreover, the pilot program emphasised the importance of cultivating bravery and resilience by encouraging students to challenge themselves to take risks, be bold, and embrace the unknown. 

Notably, student resilience showed the most significant growth as displayed in the AGRS results, alongside the development of leadership skills, adaptability, tolerance, self-confidence, and self-reliance. Correspondingly, the program was instrumental in enhancing students’ gratitude and appreciation for their privileges, such as their family, their home lives, and the natural world. Lastly, while some concerns were raised about academic studies, overall, the benefits of disconnecting from technology and reintegrating into the real world were evident for all involved. Taken as a whole, the program fostered a sense of belonging, resilience, bravery, and promoted positive wellbeing for the students. 

The following salient themes emerged from the research:I.**Sense of belonging**

There was an improvement in students’ sense of belonging as a result of engaging in activities and overall participation in the program. It included the importance of making friends from different friendship groups and building social capacity. 

II.
**Bravery and resilience**


There was an increased sense of students feeling brave and a willingness to take risks. Their courage was observed as they stepped into the unknown, embraced new experiences, and faced their fears. Resilience showed the greatest growth, along with developing leadership skills, becoming adaptable, increasing tolerance, and enhancing self-confidence. 

III.
**Gratitude, appreciation, privilege**


The EOtC experience enhanced student gratitude and appreciation for the opportunities they are afforded, such as their family, home, and privileges. Participants also articulated a greater appreciation of the natural world and became cognisant of their privilege to undertake this unique educational experience. As this study was conducted at an elite private school, results cannot be generalised to other populations that do not have the means to experience this level of privilege.

IV.
**Transitions: positive and negative**


This theme explores the various transitions that the students experienced through their participation in the program, both positive and negative. Although there was some apprehension from students, parents, and teachers about their academic studies, the benefits of unplugging from technology were visible for all. The time away from technology was widely appreciated as a significant benefit for all parties involved. 

While some students, parents, and teachers expressed concerns about missing academics, the program provided a valuable opportunity for rebooting, unplugging from technology, and reconnecting with their daily world. The time away from technology was widely appreciated as a significant benefit for all parties involved.

Questions have been raised by several researchers about how to address the privileged space of residential EOtC programs ([62]; [64]; [73]). Lack of equity in EOtC is also apparent in the perpetuation of hidden messaging that generally signals rigid cultural ideals, as discussed by [37] ([37]). This study does not attempt to address these barriers but merely highlights them for future research.

As mentioned by [10] ([10]), a program designed for females and led by females would be the ideal way forward. Elements of social role theory ([38]) and gendered leadership ([21]; [22]) play a pivotal role in this milieu to ensure girls are surrounded by role models who demonstrate effective leadership ([33]; [73]). This would be ideal for future iterations of this EOtC program.

## 4. Conclusions

This study investigated the role of ‘education outside the classroom’ (EOtC) in fostering a sense of belonging, resilience, and wellbeing among Year 9 female students in an Australian secondary school. Two cohorts participated in a four-week program, with mixed-methods research capturing quantitative data from surveys and qualitative insights from focus groups and interviews. Results indicated improvements in students’ sense of belonging and resilience, surpassing national averages. However, scores were high to begin with, likely due to socio-economic factors. While academic buoyancy scores were lower than the national baseline, a modest increase was noted post-program. The findings demonstrate that EOtC is a valuable approach for holistic student development and wellbeing enhancement. 

## Figures and Tables

**Figure 1 behavsci-15-01010-f001:**
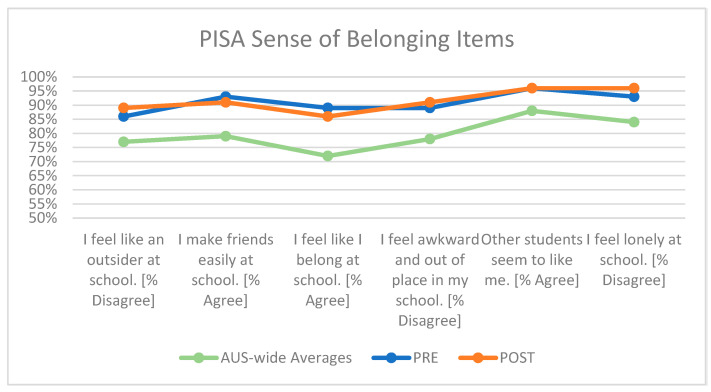
Wave 1 PISA ‘Sense of Belonging’ mean score differences, pre-/post-survey.

**Figure 2 behavsci-15-01010-f002:**
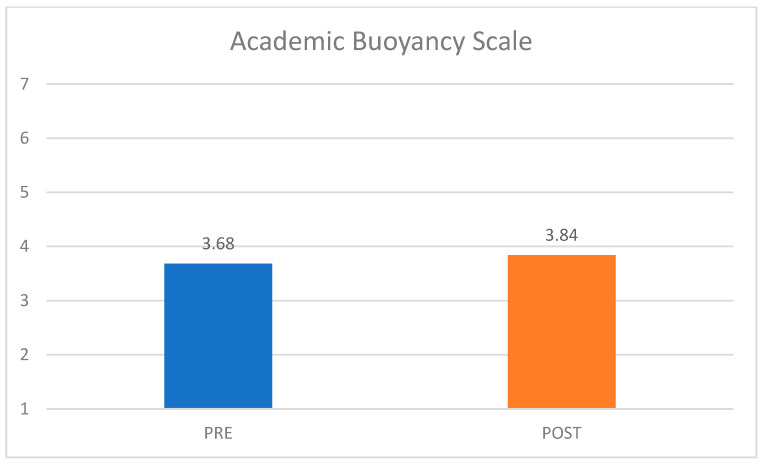
Wave 1 Academic Buoyancy Scale mean score differences, pre-/post-survey.

**Figure 3 behavsci-15-01010-f003:**
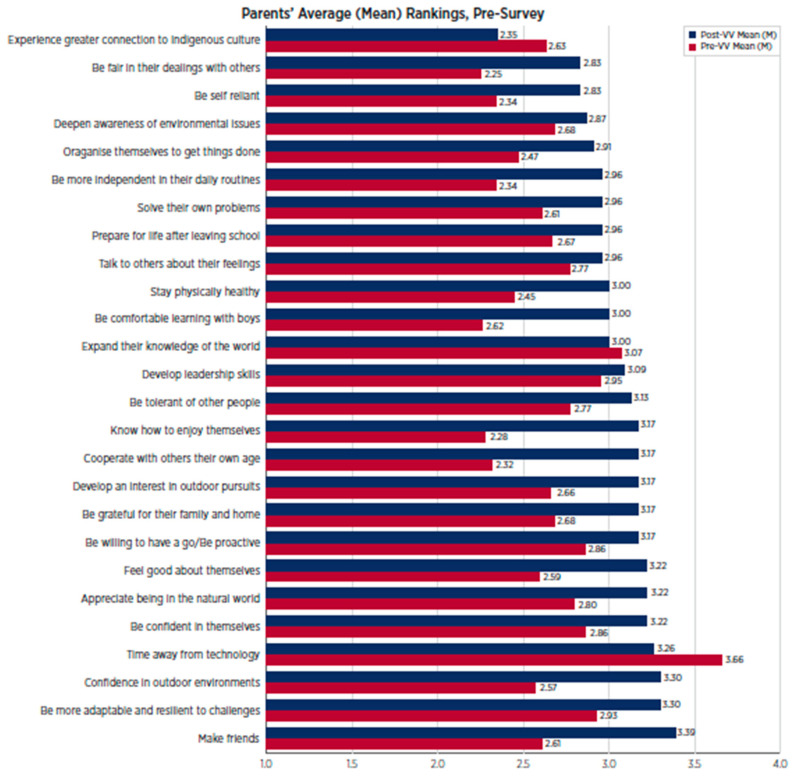
Wave 1 parents’ ranking of potential outcomes for their child pre- and post-program.

**Figure 4 behavsci-15-01010-f004:**
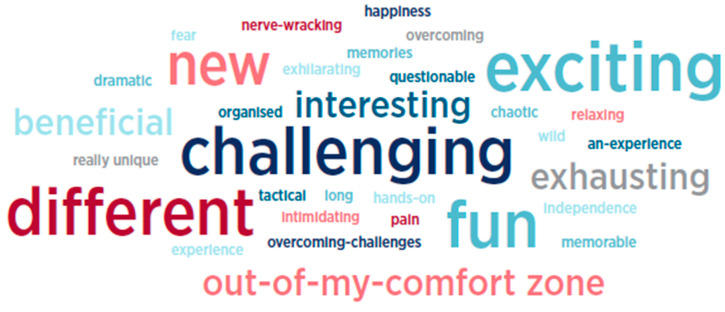
Three summative words—Wave 1 students.

**Figure 5 behavsci-15-01010-f005:**
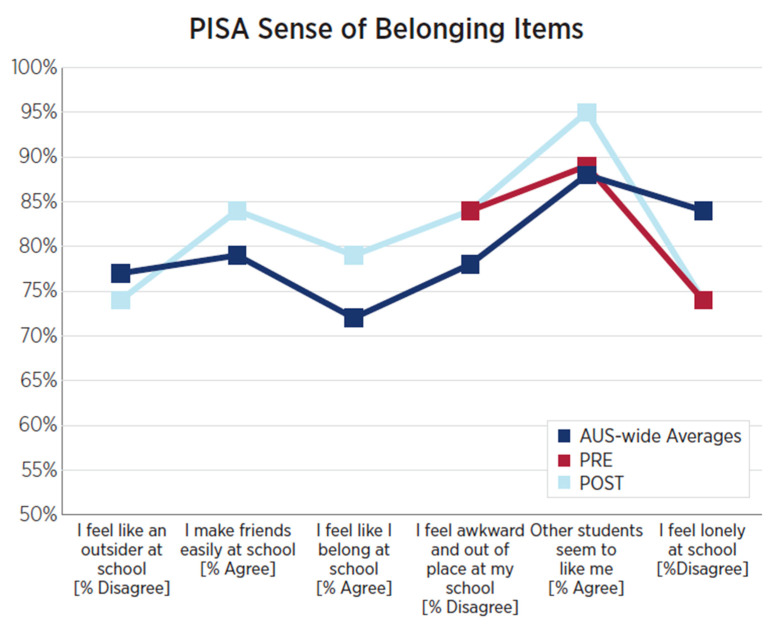
Wave 2 PISA ‘Sense of Belonging’ mean score differences, pre- and post-survey.

**Figure 6 behavsci-15-01010-f006:**
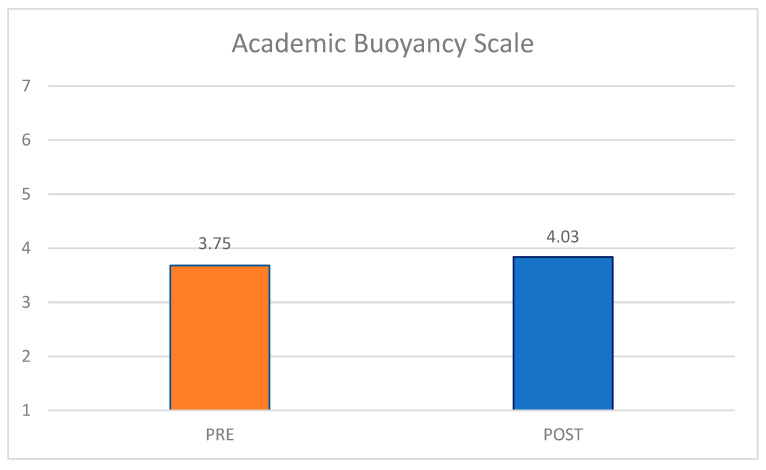
Wave 2 Academic Buoyancy Scale mean score differences, pre-/post-survey.

**Figure 7 behavsci-15-01010-f007:**
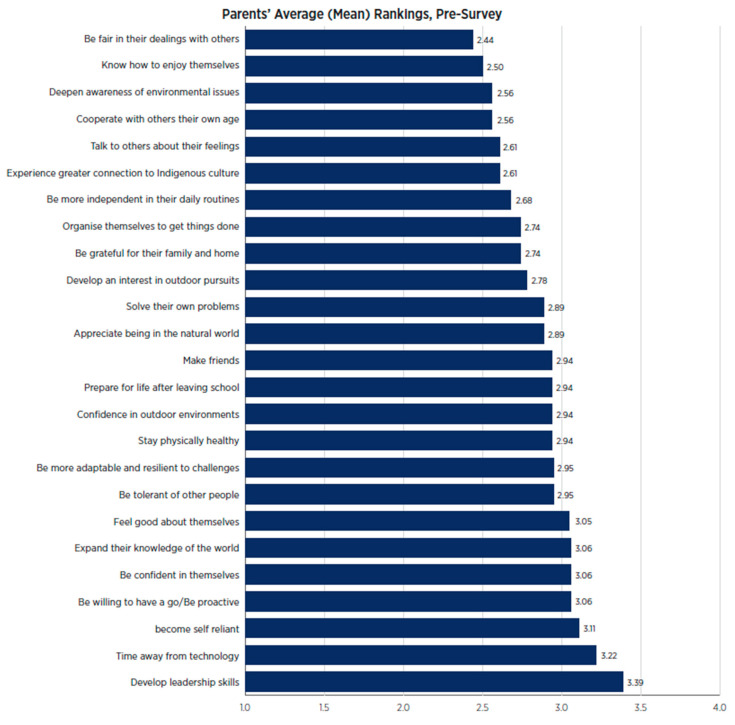
Wave 2 pre-program parent survey data.

**Figure 8 behavsci-15-01010-f008:**
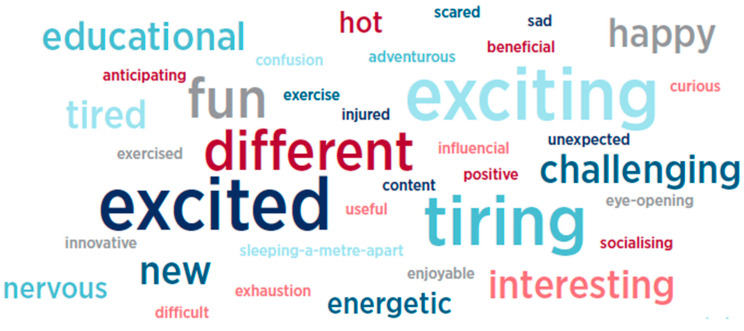
Wave 2—three words.

**Table 1 behavsci-15-01010-t001:** Research methods implemented.

Population	Quantitative Methods	Number	Qualitative Methods	Number
Year 9 student participants	Student pre-program survey	2 Surveys (Wave 1 and 2)	Individual audio interviews	63 interviews
	Student post-program survey	2 Surveys (Wave 1 and 2)	Focus groups	25 focus groups
Parent/s (of above student cohort)	Parent pre-program survey	2 Surveys (Wave 1 and 2)	Individual audio interviews	4 interviews
	Parent post-program survey	1 Survey (Wave 1 only)	N/A	
EOtC campus program staff	N/A		Individual audio interviews	3 interviews
Other school staff	N/A		Individual audio interviews	9 interviews

**Table 2 behavsci-15-01010-t002:** Paired samples mean comparison, AGRS subscales.

	Pre-Survey			Post-Survey			*t*-Test (*df*)	*p*	Cohen’s *d*
	*n*	*M*	*SD*	*n*	*M*	*SD*			
Approach to Challenges	44	3.60	0.45	44	3.73	0.51	−2.604 (43)	0.006	0.27
Self-Efficacy	44	3.75	0.44	44	3.84	0.47	−1.559 (43)	0.063	0.20
Relationship Building	44	3.87	0.51	44	3.91	0.52	−0.838 (44)	0.203	0.08

**Table 3 behavsci-15-01010-t003:** Paired samples mean comparison, AGRS subscales pre-survey.

	Pre-Survey			Post-Survey			*t*-Test (*df*)	*p*	Cohen’s *d*
	*n*	*M*	*SD*	*n*	*M*	*SD*			
Approach to Challenges	19	3.52	0.45	19	3.64	0.54	−1.701 (18)	0.053	0.24
Self-Efficacy	19	3.70	0.52	19	3.68	0.56	−1.615 (18)	0.273	0.04
Relationship Building	19	3.75	0.44	19	3.87	0.46	−1.662 (18)	0.057	0.27

**Table 4 behavsci-15-01010-t004:** Program differences between cohorts 1 and 2.

Wave 1	Wave 2
56 girls	28 girls
Co-educational	Single sex
First-ever cohort with no expectations	Expectations established from first cohort

## Data Availability

The data sets presented in this article are not available because this data concerns young children, and parents did not consent to the inclusion of their children’s data in a data set that may be shared with other researchers.

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
