# Peer review of "Wellbeing, Sense of Belonging, Resilience, and Academic Buoyancy Impacts of Education Outside the Classroom: An Australian Case Study"

_behavsci, 2025, doi:10.3390/bs15081010_

Round 1

Reviewer 1 Report

Comments and Suggestions for Authors

Overall, an interesting study although some work is needed to more directly tie the research data to the findings section. As well, some of the graphs suggest a ceiling effect (the original numbers were already high, and stayed high) which may need to be addressed. The quantitative data should be better described, including the items measured by each survey tool (this could be achieved through an appendix). The qualitative data collected appears to only be the 3-word summaries, with no explanation of what data was provided directly by the focus groups and interviews or how this data was analyzed. A more clear through line between the data analysis and findings needs to be articulated in the discussion section. The context and description of the outdoor experiences should be in the introduction or early in the methods section. The study was described as a pilot study at the end of the methods and limitation sections, but that should also be included in the abstract and introduction, with a description of why this study is a pilot study.

Comments on the Quality of English Language

While the writing is clear overall, some ideas are not well described or are over-generalized, particularly in the data analysis and findings sections. Review the manuscript for flow and coherence at both the sentence, paragraph, and section levels. Some inconsistency of font across different sections, and given the amount of data collected, the focus of this paper could be either the quantitative or qualitative findings rather than trying to do both.

Author Response

Thank you for your comments. Please see below our responses to comments. 

Response 

Overall, an interesting study although some work is needed to more directly tie the research data to the findings section.  

Thank you. A clearer through line has been established that ties methods, results and discussion together more cohesively. 

As well, some of the graphs suggest a ceiling effect (the original numbers were already high, and stayed high) which may need to be addressed.  

Thank you for your comment. This has now been addressed in the discussion relating to student socio-economic background.   

The quantitative data should be better described, including the items measured by each survey tool (this could be achieved through an appendix).  

This is covered in the methodology, with each instrument explicitly outlined.  

The qualitative data collected appears to only be the 3-word summaries, with no explanation of what data was provided directly by the focus groups and interviews or how this data was analyzed.  

Thank you. This has been expanded on in Methods and Results sections which link more effectively to the discussion.  

A more clear through line between the data analysis and findings needs to be articulated in the discussion section.  

Clearer links have been made from the results to the discussion.  

The context and description of the outdoor experiences should be in the introduction or early in the methods section.  

Context and description arenow in the introduction.  

The study was described as a pilot study at the end of the methods and limitation sections, but that should also be included in the abstract and introduction, with a description of why this study is a pilot study. 

This is now addressed in the research objectives in the introduction, with a description of why this was a pilot study.  

Reviewer 2 Report

Comments and Suggestions for Authors

The paper rethinks the issue of outdoor education and separation from technology with a strong contemporary relevance, even if temporary. The author, now a reviewer, comes from the Italian context, which has been at the forefront of open classrooms and free schools since the 1970s. Certainly, rethinking education outside the classroom, with the acronym EOtC, allows us to retrace the steps that have helped people become critical, which is sorely lacking today. This contribution is of great value for the attention it pays to the state of the art and for the contribution it offers. Unfortunately, we find ourselves in an era of sad passions, unrest, violence, and clubs for men. We must address this globally dangerous situation with progressive educational practices dedicated to inclusion, critical thinking, and compassion for others. Thank you for this research and enrichment educational approach. Keep up the good work.

Author Response

Comment  

Response 

The paper rethinks the issue of outdoor education and separation from technology with a strong contemporary relevance, even if temporary.  

Thanks 

The author, now a reviewer, comes from the Italian context, which has been at the forefront of open classrooms and free schools since the 1970s.  

Excellent, you have extensive knowledge in the field and we welcome your insights. 

Certainly, rethinking education outside the classroom, with the acronym EOtC, allows us to retrace the steps that have helped people become critical, which is sorely lacking today.  

Agree wholeheartedly  

This contribution is of great value for the attention it pays to the state of the art and for the contribution it offers.  

Thanks for your glowing reviews. 

Unfortunately, we find ourselves in an era of sad passions, unrest, violence, and clubs for men.  

We unanimously concur. 

We must address this globally dangerous situation with progressive educational practices dedicated to inclusion, critical thinking, and compassion for others.  

No argument form this writing team. Appreciate your stance. 

Round 2

Reviewer 1 Report

Comments and Suggestions for Authors

Thank you for the careful attention to my previous comments. The updated draft is well done and comprehensive.